# Evaluation of Three Formulations of Essential Oils in Broiler Chickens under Cyclic Heat Stress

**DOI:** 10.3390/ani11041084

**Published:** 2021-04-10

**Authors:** Jared Ruff, Guillermo Tellez, Aaron J. Forga, Roberto Señas-Cuesta, Christine N. Vuong, Elizabeth S. Greene, Xochitl Hernandez-Velasco, Álvaro J. Uribe, Blanca C. Martínez, Jaime A. Angel-Isaza, Sami Dridi, Clay J. Maynard, Casey M. Owens, Billy M. Hargis, Guillermo Tellez-Isaias

**Affiliations:** 1Department of Poultry Science, University of Arkansas, Fayetteville, AR 72701, USA; jaruff@uark.edu (J.R.); memotellez98@gmail.com (G.T.J.); ajforga@uark.edu (A.J.F.); rsenascu@uark.edu (R.S.-C.); vuong@uark.edu (C.N.V.); esgreene@uark.edu (E.S.G.); dridi@uark.edu (S.D.); cjm019@uark.edu (C.J.M.); cmowens@uark.edu (C.M.O.); bhargis@uark.edu (B.M.H.); 2Departamento de Medicina y Zootecnia de Aves, Facultad de Medicina Veterinaria y Zootecnia, UNAM, 04510 Cd. de Mexico, Mexico; xochitl_h@yahoo.com; 3Promitec, Bucaramanga, Santander, Colombia; gerencia@promitec.com.co (Á.J.U.); innovacion@promitec.com.co (B.C.M.); nutricionanimal@promitec.com.co (J.A.A.-I.)

**Keywords:** broiler chickens, essential oils, heat stress, *Lippia origanoides*, *Rosmarinus officinalis*

## Abstract

**Simple Summary:**

The use of essential oils in animal feeding has been practiced for their role as antibacterial, antiviral, antifungal, antioxidant, digestive stimulants, immunomodulators, hypolipidemic agents, and heat stress alleviators. The objective of the present research was to assess dietary supplementation of three formulations of essential oils on performance, bone mineralization, carcass component weights, intestinal permeability, anti-inflammatory, and antioxidant biomarkers in broiler chickens under cyclic heat stress. Heat stress reduced the body weight, feed intake, and bone strength through bone mineralization, while increasing feed conversion, gut permeability, IFN-γ and IgA levels in serum when compared with thermoneutral control broilers. Interestingly, in the present study, all three essential oils treatments partially mitigated these harmful effects at statistically significant levels compared with heat stress control chickens. These results suggest that the strategic use of some essential oils formulations during periods of stress, such as cyclic heat stress, could ameliorate adverse effects.

**Abstract:**

The objective of the present research was to assess the dietary supplementation of three formulations of essential oils (EO) in chickens under heat stress (HS). Day-of-hatch Cobb 500 chicks (*n* = 500) were randomly distributed into four groups: 1. HS control + control diets; 2. HS + control diets supplemented with 37 ppm EO of *Lippia origanoides* (LO); 3. HS + control diets supplemented with 45 ppm LO + 45 ppm EO of *Rosmarinus officinalis* (RO) + 300 ppm red beetroot; 4. HS + 45 ppm LO + 45 ppm RO + 300 ppm natural betaine. Chickens that received the EO showed significant (*p* < 0.05) improvement on BW, BWG, FI, and FCR compared to control HS chickens. Average body core temperature in group 3 and group 4 was significantly (*p* < 0.05) reduced compared with the HS control group and group 2. Experimental groups showed a significant reduction in FITC-d at 42 days, a significant increase in SOD at both days but a significant reduction of IFN-γ and IgA compared with HS control (*p* < 0.05). Bone mineralization was significantly improved by EO treatments (*p* < 0.05). Together these data suggest that supplemental dietary EO may reduce the harmful effects of HS.

## 1. Introduction

The gastrointestinal tract (GIT) is the largest lymphoid organ containing an estimated 70–80% of the immune cells [1]. At the same time, it harbors a microbial population that is ten times larger than the number of somatic cells and their genes [2]. In the intestine, consisting of a single line of epithelial cells, enterocytes are linked to each other by several complexes of tight junction proteins [3]. The GIT is one of the most sensitive organs to stress and inflammation. Disruption of this fragile organ increases leakage of bacteria and antigens to the bloodstream, which can result in severe chronic systemic inflammation [4,5,6].

Intensification in heat temperatures has been recorded in the last two decades, and these increments are anticipated to continue [7]. These environmental changes have severe repercussions in all forms of life on earth [8,9]. In recent years, HS has become one of the significant economic concerns for the poultry industry [10]. In HS, animals take up more heat than is diffused. To compensate for the absence of sweat glands, chickens use other methods to regulate body temperature, such as convection, evaporation, and radiation [11]. However, under commercial conditions, HS has a devastating effect on broiler chicken homeostasis [12]. The detrimental impact of HS can vary from heat fatigue and performance reduction to cellular, tissue, and organ injuries resulting in death [10,12,13]. Under such challenging conditions, chickens undergo profound physiological and behavioral changes as they struggle to return to homeostasis. 

Furthermore, chickens suffering from chronic heat stress develop an increased heterophil: lymphocyte ratio, which is a reliable stress indicator [14]. In the United States, annual losses due to HS have been estimated at 2.36 billion dollars due to HS [10].

As with any other chronic stress condition (regardless of its origin, e.g., chemical, biological, physical, nutritional, or emotional), reproduction and growth functions are shutdown while the survival conditions are activated [12]. Moreover, the excess free radicals produced under chronic stress, such as reactive oxygen, nitrogen or chlorine species, often exceed the antioxidant capacity of vital enzymes such as catalase, superoxide dismutase and glutathione peroxidase [15,16]. Perhaps the cellular molecules that suffer immediate repercussions are the cellular membrane lipid and protein components and the mitochondrial membrane by lipid peroxidation. Damage in those structures compromises the whole cellular physiology of tissues and organs. One of the cellular mechanisms to revert oxidative stress is the production of several heat shock proteins (HSP) that repair damage proteins and regulate apoptosis [6,17].

On the other hand, the increased awareness and concern over the antibiotic residues as well as the emergence of antibiotic-resistant bacteria has created the necessity of replacing antibiotic growth promoters (AGPs) with other products like phytogenics, prebiotics, probiotics, organic acid, enzymes, and vaccines. Phytogenics are a group of natural growth promoters used as feed additives, derived from herbs, spices or other plants. 

Essential oils (EO) represent a concentrated form of phytogenics, containing mainly the active ingredients of the plants as secondary metabolites. Several studies have shown the antibacterial, antiviral, antifungal, antioxidant, digestive stimulants, immunomodulators, hypolipidemic agents, and heat stress alleviators of EO such as carvacrol and thymol, which are present in high concentrations in oregano and rosemary [18,19,20]. 

Other beneficial effects of EO include appetite stimulation, improvement of enzyme secretion related to food digestion, and immune response modulation [21]. Some EO are generally used as a blend with a carrier oil or in combination with other plants rich in betaine, a central amino acid that plays important metabolic and osmolar activities in the feed to enhance chickens’ productive performance [22,23,24]. Hence, EO had been successfully used as a dietary antibiotic replacer without residues [25]. The antioxidant effect of EO has beneficial effects in situ, and it has been reported that this antioxidant property also increases the shelf life of the feed as well as the meat from the chickens fed with EO by reducing lipid peroxidation [26]. Moreover, the lean meat produced by chickens fed with EO reduces the risk of hyperlipidemia in the consumers [27]. 

However, the spectrum of phytogenic feed additives is vast and does not only consist of essential oils, but also includes other active ingredient groups, such as pungent substances, bitter substances, saponins, flavonoids, mucilages and tannins [28,29].

Betaine, the trimethyl derivative of glycine, the simplest and crucial proteinogenic amino acid [30], is produced by many plants and metazoans. Betaine works in the methylation processes, hence substituting other methyl group donors like choline and methionine [31]. There is increasing evidence that betaine is a highly valuable feed additive in many farm animals [23]. Betaine has also been shown to protect cells from osmotic stress and allows cells to continue regular metabolic activities in conditions that would generally inactivate the cell [24]. Furthermore, betaine donates its labile methyl group, which can be used in transmethylation reactions to synthesize substances like carnitine; and thus, affect animal fat metabolism [32]. Because of these reasons, betaine is also known as the "carcass modifier" as it improves carcass quality and yield, particularly under HS conditions, due to its osmoprotective and osmoregulatory role in cells [33,34].

All the mention properties are encouraging researchers to evaluate EO as an alternative to AGPs [22,35]. Hence, it is not surprising that EO are playing an interesting use as feed additives for monogastric and ruminants [21]. In the present study, three formulation of EO are assessed. One formulation is a commercial source of *Lippia origanoides* (Natbio Esencial Premix^®^, Bucaramanga, Colombia). The other two formulations are experimental, and both contain EO from *Lippia origanoides* and *Rosmarinus officinalis.* Both experimental formulations contain betaine. However, the source of betaine differs in its origin. In one formulation, the source of betaine comes from beetroot extract (*Beta vulgaris*) while in the other comes from a mix of several plant extracts (AMORVET, Bhagwanpur, India). Therefore, the objectives of the present research were to confirm and evaluate the antioxidant and anti-inflammatory properties of dietary supplementation of three formulations of essential oils (EO) on performance, bone mineralization, carcass component weights, intestinal permeability, anti-inflammatory, and antioxidant biomarkers in broiler chickens under cyclic heat stress.

## 2. Materials and Methods

### 2.1. Ethics

All animal handling procedures complied with the Institutional Animal Care and Use Committee (IACUC) at the University of Arkansas, Fayetteville. Explicitly, the IACUC approved this study under protocol # 18030.

### 2.2. Essential Oil Products

Three formulations of EO were provided by Promitec Santander S.A. and feed inclusion based on the manufacturer’s recommendations. One formulation was commercial, and the other two experimental formulations. The formulations were: Formula 1:37 ppm EO of *Lippia origanoides*, with feed inclusion of 300 g/ton of feed (Natbio Esencial Premix^®^, Bucaramanga, Colombia); Formula 2: 45 ppm EO of *Lippia origanoides* + 45 ppm EO of *Rosmarinus officinalis* + 300 ppm of beetroot extract (*Beta vulgaris*), with feed inclusion of 700 g/ton of feed; Formula 3: 45 ppm EO of *Lippia origanoides* + 45 ppm EO of *Rosmarinus officinalis* + 300 ppm of Natural Betaine (AMORVET, Bhagwanpur, India), with feed inclusion of 700 g/ton of feed.

Due to EO’s rapid absorption and metabolism by enterocytes, encapsulation of these feed additives to augment their efficiency has been proposed [36]. In the present study, all EO products evaluated were microencapsulated with maltodextrin by spray drying to improve the encapsulation efficiency and bioavailability and extend the EO’s shelf life. 

### 2.3. Facilities and Experimental Design

Experimental design is summarized in Figure 1. This study was conducted at the Poultry Experimental Research Laboratory (PERL) facility at the University of Arkansas during late summer (July–August), which includes individual environmentally controlled rooms equipped with their own air conditioning unit and digital thermostat to control temperature. Day-of-hatch Cobb 500 male broiler chicks (*n* = 500) were obtained from a commercial hatchery. Upon arrival, all chickens were vaccinated with a coccidia vaccine (Coccivac^®^-B52, Merck Animal Health, Maxton, NC, USA), neck tagged, and randomly distributed into four groups. Group 1: HS control + control diets; Group 2: HS + control diets supplemented with 37 ppm EO of *Lippia origanoides* (LO); Group 3: HS + control diets supplemented with 45 ppm LO + 45 ppm EO of *Rosmarinus officinalis* (RO) + 300 ppm beetroot; Group 4: HS + 45 ppm LO + 45 ppm RO + 300 ppm Natural Betaine. 

The starter, grower, and finisher diets used in this research were adjusted to breeder’s recommendations [37] and formulated to provide an adequate supply of nutrients (Table 1). No growth promoters or coccidiostats were included in the diets. Groups were allocated to ten environmental rooms. Each room was divided into two pens (150 × 300 cm), each containing separate feeders and watering systems, five replicates per treatment with 25 birds/replicate (*n* = 125). At placement, chickens were exposed to 34 °C and relative humidity at 55 ± 5% for the first 7 d. During cyclic HS, chickens received 35 °C for 12 h daily from day 7 to day 42. Relative humidity remained constant at 55 ± 5%. On d 18, eight chickens were randomly selected to orally insert a Thermochron temperature logger (iButton, DS1922L, Embedded Data Systems, Lawrenceburg, KY, USA). The devices stayed in the gizzard for measurement of body temperature as described by Flees et al. [38]. Every minute during the first two hours after initiation of heat stress, and every subsequent hour, the chickens’ body temperature was logged (*n* = 1.254). 

Performance parameters, body weight (BW), body weight gain (BWG), feed intake (FI), and feed conversion ratio (FCR) were evaluated at d 21 and 42. On the same days, four random chickens per pen were selected (*n* = 20) and orally gavaged with 8.32 mg/kg of body weight of fluorescein isothiocyanate-dextran (FITC-d, MW 3–5 KDa; St. Louis, MO, USA). One hour after FITC-d administration, chickens were humanely euthanized by CO_2_ inhalation. Blood samples were collected from the femoral vein and centrifuged (1000× *g* for 15 min) to separate the serum.

### 2.4. Serum Levels of Fluorescein Isothiocyanate-Dextran

Serum levels of FITC-d were used as a biomarker to evaluate intestinal permeability as described by Baxter et al. [39]. 

### 2.5. Superoxide Dismutase Activity

Superoxide dismutase (SOD) activity was measured in serum samples using a commercial assay kit (Cayman chemical company, Item No. 706002, Ann Arbor, MI, United States) following the manufacturer’s instructions. Three types of SOD (Cu/Zn, Mn, and FeSOD) were determined and the optimal dilution to quantify the SOD activity was 1:5. Samples were measured at 450 nm using an ELISA plate reader (Synergy HT, multimode microplate reader, BioTek Instruments, Inc., Winooski, VT, United States).

### 2.6. Serum Levels of Gamma Interferon

Gamma interferon (IFN-γ) serum levels were using a commercial assay kit from Invitrogen Corporation (Frederick, MD, USA). 

### 2.7. Serum Total Immunoglobulin A

Levels of total IgA serum levels were determined as previously described [40]. A commercial indirect enzyme-linked immunosorbent assay (ELISA) set was used to quantify IgA according to the manufacturer’s instructions (Catalog No. E30-103, Bethyl Laboratories, Inc., Montgomery, TX, USA).

### 2.8. Bone Strength

The left and right tibias from each sampled chicken on days 21 and 42 were removed to assess break strength (kg) and total ash based on fat-free tibia (%) as described by Gautier et al. [41]. Tibial diaphysis from individual birds was cleaned of adherent tissues, the periosteum was removed, and the biomechanical strength of each bone was measured using an Instron 4502 material testing machine (Norwood, MA, USA) with a 509 kg load cell. The bones were held in identical positions and the mid-diaphyseal diameter of the tibial midshaft, which was also the site of impact, was measured using a dial caliper. The maximum load at failure was determined in the tibial midsection between epiphyses, using a three-point flexural bend fixture with a total distance of 30 mm between the two lower supporting ends. The load, defined as the force in kilograms per square millimeter of cross-sectional area (kg/mm^2^), represents bone strength. The rate of loading was kept constant at 20 mm/min collecting 10 data points per second. The data were automatically calculated using Instron’s Series IX Software (Norwood, MA, USA).

### 2.9. Processing Parameters

On day 42, nine chickens per replicate pen (*n* = 45 HS) per group were selected to evaluate processing parameters. Chickens were commercially processed at the University of Arkansas Pilot Processing Plant. Because the plant was on-site, broilers did not undergo extended transportation. Grouped birds were transported on the back of a flatbed trailer to the abattoir where processing could commence. Feed was withheld for ten h before slaughter, and broilers were weighed individually at the plant. Automated equipment was used for electrical stunning, scalding, picking, vent opening, and evisceration. Birds were scalded at 53 °C for 120 s. Carcasses were prechilled at 12 °C for 15 min and chilled (immersion) at 1 °C for 2.75 h. After being chilled, carcasses were drained of water and chilled weight was recorded before deboning into the subsequent parts of breast, tender, wing, whole leg, and rack and their respective weights were recorded.

### 2.10. Statistical Analysis

All data were subjected to analysis of variance (ANOVA) as a completely randomized design using the General Linear Models procedure of SAS [42]. Significant differences among the means were determined by Duncan’s multiple range test at *p* < 0.05.

## 3. Results

The results of the evaluation of EO on broiler chickens exposed to cyclic HS on performance parameters and carcass component weights are summarized in Table 2. Chickens that received the EO showed significant (*p* < 0.05) improvement on BW, BWG, FI, and FCR compared to control HS chickens (Table 2). Cyclic HS reduced all parameters evaluated for carcass component weights (hot carcass, chilled carcass, wing, breast, tender, and leg and quarter). Interestingly, in the present study, the formulations of EO in group 3 (supplemented with 45 ppm *Lippia origanoides* + 45 ppm Rosmarinus officinalis + 300 ppm beetroot) and group 4 (supplemented 45 ppm *Lippia origanoides* + 45 ppm *Rosmarinus officinalis* + 300 ppm natural betaine) significantly mitigated the harmful effects of HS in carcass component weights when compared with the HS control chickens (Table 2).

The evaluation of essential oils on broiler chickens exposed to cyclic heat stress on body core temperature, serum biomarkers for intestinal inflammation, and bone parameters at days 21 and 42 is summarized in Table 3. Only two hours after introducing HS in the experimental groups, a significant (*p* < 0.05) increase in the body core temperature of the chickens was observed and heightened body temperature during heat stress was observed through the trial (data not shown). Average body core temperature in group 3 (supplemented with 45 ppm *Lippia origanoides* + 45 ppm *Rosmarinus officinalis* + 300 ppm beetroot) and group 4 (supplemented 45 ppm *Lippia origanoides* + 45 *ppm Rosmarinus officinalis* + 300 ppm natural betaine) was significant reduced compared with HS control group and group 2, supplemented with 37 ppm EO of *Lippia origanoides* (Table 3). At 21 days, only groups 3 and 4 showed a significant reduction in serum FITC-d, intestinal permeability biomarker. However, all experimental groups treated showed a significant reduction in FITC-d at 42 days compared with control HS chickens. In the present study, experimental treated chickens had a significant increase in serum concentrations of SOD at both days of evaluation compared to control HS chickens, but significant reduction in serum levels of gamma interferon and IgA (Table 3). All three experimental groups showed a significant increase in tibia break strength at both days of evaluation, however, total ash from tibia was significantly higher in groups 3 and 4 at 21 and 42 days of evaluation (Table 3).

## 4. Discussion

Thymol and carvacrol are volatile aromatic EO found in high concentrations in thyme, oregano, and rosemary. Chemically, they are secondary metabolites commonly composed of terpenoids and phenylpropanoids [29]. As feed additives, these EO have been shown to enhance nutrient bioavailability, productive and reproductive performances [19]. In the present study, groups that received EO and were exposed to cyclic HS exhibited improved BW and BWG compared with HS control chickens at 21 d and 42 d, as well as FCR (d 42 only). These results agree with other studies that have shown chickens under HS that received EO from oregano [43], rosemary [44], or betaine [26] all had performance improvements. Other studies suggest that the increased performance observed in chickens supplemented within EO is due to stabilization of the microbial eubiosis in the gut, increased digestive enzyme secretion and stimulated appetite [19,22].

Heat stress has profound metabolic and physiological effects in modern broiler chickens, such as downregulated gene expression of lipoprotein lipase and hepatic triacylglycerol lipase and upregulation of adipose triglyceride lipase [38]. As a result, these gene expression changes are associated with an increase in abdominal, intermuscular, and subcutaneous tissue fat deposition in chickens affected by HS. Additionally, HS induces cellular osmotic and dehydration associated with increased plasma triglyceride and glucose, serum calcium (due to bone demineralization), and total serum protein. These alternations have significant implications in the water holding capacity of the chicken meat products [45,46,47]. 

Furthermore, chronic HS increases body temperature and respiration rate, instigating respiratory alkalosis. All the above changes directly affect chicken meat quality by reducing breast muscle water content and color, while also increasing carbonyl concentrations and thiobarbituric acid reactive substances (TBARS) formed as a byproduct of lipid peroxidation [12,48,49]. Other investigators have reported the detrimental effects of chronic HS on carcass yield and meat quality [26,33,50,51,52]. Interestingly, in the present study, chickens in groups 3 and 4 showed significant improvements in carcass component weights compared with HS control chickens. Both groups included the combination of EO and a source of the crucial proteinogenic amino acid betaine, also known as the "carcass modifier", due to its osmoprotective and osmoregulatory properties in cells, especially under HS conditions [33,34].

Maintenance of optimal bird health is critical for improved withstanding against the physiological challenges associated with HS. Various managerial and nutritional strategies have been proposed to mitigate the adverse effects of HS in chickens, with plant-based additives showing promise [18,19]. In this regard, EO have received particular attention as natural alternatives for replacing AGPs in poultry diets due to their role as antibacterial, antiviral, antifungal, antioxidant, immunomodulatory, hypolipidemic agents, and heat stress alleviators [20,21].

Our results confirmed previous reports indicating that HS compromised the intestinal barrier, increasing gut permeability [53]. On day 21 of evaluation, EO formulations included in groups 3 and 4 showed a significant reduction in serum FITC-d compared to the rest of the groups exposed to HS. However, at 42 d of evaluation, all three formulations of EO in chickens exposed to cyclic HS significantly reduced leakage of FITC-d when compared with control HS chickens. FITC-d is a large molecule (3–5 kDa) that does not usually leak through the intact gastrointestinal tract barrier. However, when conditions disrupt the tight junctions between epithelial cells, the FITC-d molecule can enter circulation, as demonstrated by an increase in trans-mucosal permeability associated with chemically induced disruption of tight junctions by elevated serum levels of FITC-d after oral administration [40]. Interestingly, a recent review indicates that EO decreases intestinal permeability by increasing the gene expression of tight junction (TJ) proteins, downregulating gene expression of proinflammatory cytokines, and increasing the proliferation of goblet cells [19].

Alternatively, perhaps the most studied EO properties are their antioxidant activity, radical scavenging, and antimicrobial capabilities [54]. Under normal temperature conditions, the antioxidant systems of chickens are in a state of dynamic equilibrium and can adapt to manage normal challenges. During HS; however, reactive oxygen species are produced at elevated levels beyond which the system can handle, resulting in oxidative stress [15]. The superoxide dismutases (SOD) are key enzymes in the conversion of superoxide free radicals into hydrogen peroxide and molecular oxygen [6,7]. In the present study, all three dietary formulations of EO tested increased serum concentrations of SOD on days 21 and 42 compared with the HS control chickens. Free radical scavenging capacity protects the integrity of cellular and mitochondrial membranes from lipid peroxidation [55,56,57]. Essential oils from rosemary, oregano, thyme, and turmeric increased the antioxidant response element in enterocytes suffering oxidative stress, suggesting a unique mechanism by these compounds [58]. Other studies have confirmed the antioxidant activity of Lippia [56,59,60,61,62], rosemary [61] and beetroot [63].

Beetroot (*Beta vulgaris*) is particularly fascinating due to the antioxidant, anti-inflammatory, and apoptosis properties of its betalain pigments [64]. More recently, the high concentration of nitrate (NO_3_−) present in beetroot has been linked to the endogenous production of nitric oxide (NO). This has been associated with involvement in vascular, inflammatory, apoptosis, and neurotransmission responses, which have received worldwide attention [65,66,67]. Interestingly, chickens exposed to cyclic HS that receive the EO formulations presented a significant reduction in serum concentrations of the proinflammatory cytokine IFN-γ. Likewise, recent studies published by our laboratory have confirmed IgA as a reliable serum biomarker to evaluate intestinal inflammation [40,68,69,70]. In the present study, serum levels of IgA at 21 and 42 days of evaluation were significantly reduced in all experimental groups compared with control HS chickens, suggesting that EO downregulates the inflammatory response of HS. In the present study, just two hours after introducing HS in the experimental groups, a significant increase in body temperature was observed, which remained throughout the trial. Nevertheless, it was noteworthy to follow that chickens in groups 3 and 4, both containing betaine, significantly reduced body core temperature compared with the other two HS groups. These findings agree with Attia et al. [71], who reported that dietary betaine improved performance parameters, rectal temperature, respiration rate, blood pH, meat quality, and humoral immune response in chickens under HS.

HS is associated with a reduction in feed intake and inflammation; these conditions have a high correlation with a reduction in bone mineralization and bone restoration [30,72,73,74,75,76]. In the present study, control chickens exposed to cyclic HS showed a significant reduction in bone mineralization as evaluated by tibia break strength and total ash from the tibia, confirming the results of a previous study [53]. However, groups 3 and 4 that received a mix of EO and betaine showed a significant improvement in bone parameters evaluated on days 21 and 42. 

In summary, the results of the present study suggest that the supplementation and combination of EO from *Lippia origanoides, Rosmarinus officinalis*, and either beetroot or natural betaine improves performance, carcass component weights, intestinal permeability, antioxidant, and anti-inflammatory properties in broiler chickens under cyclic HS. Studies to evaluate these properties and the bactericidal activities against *Clostridium perfringens* in a necrotic enteritis laboratory model are currently being evaluated.

## 5. Conclusions

Heat stress reduced the performance parameters of BW, feed intake, and bone strength through bone mineralization, while increasing feed conversion, gut permeability, IFN-γ, and IgA levels when compared with thermoneutral control broilers. However, strategic use of EO and betaine during a period of stress, such as heat stress, could help to reduce the negative effects in broiler chickens.

## Figures and Tables

**Figure 1 animals-11-01084-f001:**
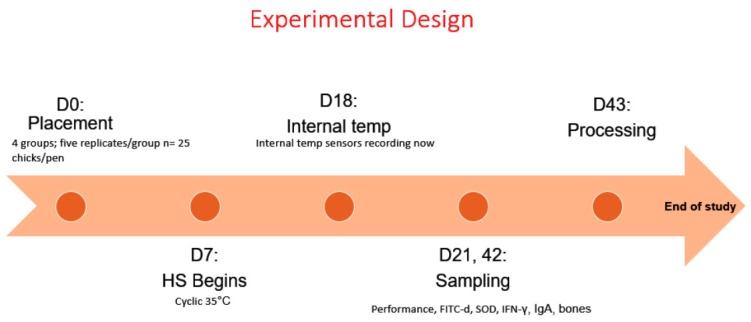
Experimental design.

**Table 1 animals-11-01084-t001:** Ingredient composition and nutrient content of the corn–soybean diet used on an as-is basis.

Feed Ingredients	Stater Phase (d 1 to 7)	Grower Phase (d 8 to 14)	Finisher Phase (d 15 to 25)
Ingredients (%)			
Corn	51.80	57.81	59.64
Soybean meal	37.66	31.62	27.23
DDGS 8.1% EE	4.00	4.00	6.00
Poultry fat	3.24	3.44	4.38
Limestone	1.01	1.06	1.03
Dicalcium phosphate	1.00	0.88	0.64
Salt	0.35	0.35	0.31
DL-methionine	0.29	0.31	0.28
L-lysine HCl	0.12	0.13	0.12
Mineral premix ^b^	0.10	0.10	0.10
Vitamin premix ^a^	0.10	0.10	0.10
L-threonine	0.08	0.09	0.09
Choline chloride	0.06	0.06	0.05
Sodium bicarbonate	0.04	0.05	0.03
Antioxidant ^c^	0.15	0.15	0.15
Total	100	100	100
Calculated analysis			
ME (kcal/kg)	3015.00	3090.00	3175.00
Ether extract (%)	5.88	6.20	7.28
Crude protein (%)	22.30	20.00	18.70
Lysine (%)	1.18	1.05	0.95
Methionine (%)	0.59	0.53	0.48
Threonine (%)	0.77	0.69	0.65
Tryptophan (%)	0.25	0.22	0.20
Total calcium (%)	0.90	0.84	0.76
Total phosphorus (%)	0.63	0.58	0.53
Available phosphorus (%)	0.45	0.42	0.38
Sodium (%)	0.20	0.20	0.18
Potassium (%)	1.06	0.94	0.87
Chloride (%)	0.27	0.28	0.25
Magnesium (%)	0.19	0.18	0.17
Copper (%)	19.20	18.46	18.85
Selenium (%)	0.28	0.27	0.26
Linoleic acid (%)	1.01	1.13	1.16

^a^ Vitamin premix supplied the following per kg: vitamin A, 20,000,000 IU; vitamin D3, 6,000,000 IU; vitamin E, 75,000 IU; vitamin K3, 9 g; thiamine, 3 g; riboflavin, 8 g; pantothenic acid, 18 g; niacin, 60 g; pyridoxine, 5 g; folic acid, 2 g; biotin, 0.2 g; cyanocobalamin, 16 mg; and ascorbic acid, 200 g (Nutra Blend LLC, Neosho, MO 64850).; ^b^ Mineral premix supplied the following per kg: manganese, 120 g; zinc, 100 g; iron, 120 g; copper, 10–15 g; iodine, 0.7 g; selenium, 0.4 g; and cobalt, 0.2 g (Nutra Blend LLC, Neosho, MO 64850); ^c^ Ethoxyquin.

**Table 2 animals-11-01084-t002:** Evaluation of essential oils on broiler chickens exposed to cyclic heat stress on performance parameters and carcass component weights at days 21 and 42.

Performance Parameter	Heat Stress Control	*Lippia origanoides* (37 ppm)	*L. origanoides **, *R. officinalis*, Beetroot	*L. origanoides **, *R. officinalis*, Natural Betaine	Pooled SEM	*p*-Value
BW, g/broiler						
d 0	43.62	43.20	43.71	43.82	0.98	0.1457
d 21	612.30 ^b^	689.41 ^a^	680.24 ^a^	695.20 ^a^	28.90	0.0002
d 42	2119.20 ^c^	2329.90 ^ab^	2242.11 ^ab^	2380.75 ^a^	125.42	0.0001
Accumulated BWG, g/broiler						
d 0 to 21	569.30 ^b^	646.41 ^a^	637.24 ^a^	652.20 ^a^	26.78	0.0001
d 0 to 42	2016.20 ^c^	2286.90 ^ab^	2199.11 ^ab^	2337.75 ^a^	119.87	0.0002
FI, g/broiler						
d 0 to 21	790.30 ^b^	910.41 ^a^	891.24 ^a^	930.20 ^a^	32.40	0.0001
d 0 to 42	4110.20 ^c^	4355.90 ^ab^	4125.11 ^bc^	4284.75 ^a^	230.56	0.0002
Accumulated FCR						
d 0 to 21	1.30	1.32	1.31	1.33	0.87	0.1689
d 0 to 42	1.94 ^a^	1.87 ^b^	1.84 ^b^	1.80 ^b^	0.92	0.0001
Carcass component weights (g) at day 42						
Live weight	2156.25 ^c^	2240.91 ^bc^	2361.15 ^ab^	2423.30 ^a^	201.36	0.0001
Hot carcass	1640.88 ^c^	1688.86 ^bc^	1788.49 ^ab^	1836.98 ^a^	187.32	0.0002
Chilled carcass	1687.38 ^b^	1731.16 ^b^	1847.51 ^a^	1885.35 ^a^	198.86	0.0001
Wing	180.85 ^c^	183.57 ^bc^	189.55 ^ab^	196.83 ^a^	20.13	0.0001
Breast	330.02 ^b^	336.30 ^b^	371.98 ^a^	380.65 ^a^	31.10	0.0001
Leg and quarter	534.40 ^c^	548.98 ^bc^	579.13 ^ab^	588.65 ^a^	42.02	0.0002

** L. origanoides* (45 ppm); *R. officinalis* (45 ppm); Beetroot (300 ppm); Natural betaine (300 ppm). Data are expressed as the mean ± SE. ^abc^ Indicates significant differences between the treatments within the rows (*p* < 0.05).

**Table 3 animals-11-01084-t003:** Evaluation of essential oils on broiler chickens exposed to cyclic heat stress on body core temperature, serum biomarkers for intestinal inflammation, and bone parameters at days 21 and 42.

Variable	Heat Stress Control	*Lippia origanoides* (37 ppm)	** L. origanoides, R. officinalis*, Beetroot	** L. origanoides, R. officinalis*, Natural Betaine	Pooled SEM	*p*-Value
Body core temperature (°C)	42.36 ^a^	42.35 ^a^	41.98 ^b^	41.98 ^b^	0.83	0.0001
Serum FITC-d (ng/mL)						
d 21	264 ^a^	288 ^a^	152 ^b^	251 ^b^	95	0.0001
d 42	245 ^a^	165 ^b^	137 ^bc^	129 ^c^	82	0.0002
SOD (U/mL)						
d 21	7.35 ^b^	8.66 ^a^	8.55 ^a^	9.01 ^a^	0.45	0.0001
d 42	8.45 ^b^	9.73 ^a^	10.05 ^a^	10.85 ^a^	0.61	0.0002
IFN-γ (pg/ml)						
d 21	134 ^a^	118 ^b^	112 ^b^	116 ^b^	17	0.0001
d 42	251 ^a^	131 ^b^	122 ^b^	133 ^b^	22	0.0002
IgA (ng/mL)						
d 21	14 ^a^	8 ^b^	9 ^b^	8 ^b^	0.38	0.0001
d 42	16 ^a^	9 ^b^	10 ^b^	9 ^b^	0.53	0.0001
Tibia break strength (kg)						
d 21	13.79 ^b^	15.69 ^a^	16.09 ^a^	15.99 ^a^	1.12	0.0001
d 42	22.37 ^b^	29.17 ^a^	30.37 ^a^	31.37 ^a^	2.05	0.0002
Total ash from tibia (%)						
d 21	50.57 ^b^	51.67 ^b^	52.67 ^a^	53.77 ^a^	0.49	0.0001
d 42	52.33 ^b^	53.34 ^ab^	54.34 ^a^	55.04 ^a^	0.31	0.0001

** L. origanoides* (45 ppm); *R. officinalis* (45 ppm); Beetroot (300 ppm); Natural betaine (300 ppm). Data are expressed as the mean ± SE. ^abc^ Indicates significant differences between the treatments within the rows (*p* < 0.05).

## Data Availability

The data presented in this study are available on request from the corresponding author.

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
