# Peer review of "Evaluation of Three Formulations of Essential Oils in Broiler Chickens under Cyclic Heat Stress"

_animals, 2021, doi:10.3390/ani11041084_

Round 1

Reviewer 1 Report

The manuscript by Ruff et al. provides preliminary data in the application of essential oils as dietary supplements to broiler chickens exposed to heat stress and it is an interesting study.

The data from this well-executed study shows the potential of the essential oils in reducing the negative effects due to heat stress.

I only had some questions for the authors to address:

  • Line 22: make sure to use the sign for gamma
  • Lines 46-48 need to be rephrased. HS + absence of sweat glands and elevated metabolism, causing severe economic losses? Provide detailed causes for the losses.
  • Why were the chicks vaccinated for coccidia and how? Provide details and reference.
  • Would there be any logistical problems in scaling-up the EO for supplementing to much larger numbers of birds? Please discuss.
  • What would be the cost implication of supplementing EO on a larger scale? Please discuss.

Author Response

Response to Reviewer 1 Comments

Dear Reviewer, #1, thank you very much for the time you have spent on reviewing our manuscript. Your comments are very valuable and helpful for revising our paper and guiding our research. We have studied those comments carefully and have made corrections, which we hope to meet with the approval. Revised portion in the new version were included and are highlighted in yellow in the reviewed manuscript. The following is our point-by-point response to reviewers’ comments:

I only had some questions for the authors to address:

  • Line 22: make sure to use the sign for gamma

Suggestion accepted.  Thank you.

  • Lines 46-48 need to be rephrased. HS + absence of sweat glands and elevated metabolism, causing severe economic losses? Provide detailed causes for the losses.

Suggestion accepted.  Text has been modified.  Thank you.

  • Why were the chicks vaccinated for coccidia and how? Provide details and reference.

Because no growth promoters or coccidiostats were included in the diets, and the experiment was conducted in floor pens, a coccidia vaccine is suggested to be administered to reduce the probabilities of coccidiosis in the trial.  Without coccidiostats or coccidian vaccine, the risk would be too high and could potentially compromise the experiment.

  • Would there be any logistical problems in scaling-up the EO for supplementing to much larger numbers of birds? Please discuss.

Actually, one formulation was commercial (Natbio Esencial Premix®), and the other two experimental formulations.  This has clarified in the text too. Thank you.

  • What would be the cost implication of supplementing EO on a larger scale? Please discuss.

The cost to treat one metric ton of chicken feed with Natbio Esencial Premix® is $5.9 US dollars.  We have included the commercial name of the product in the text.  Thank you.

Reviewer 2 Report

The paper is a good read and congratulations to the Authors on the idea and implementation. Heat stress is a common problem in the broiler industry, because heavy-weight chickens do not endure heat waves and mortality rate is high due to heat. For this reason, it is important to test and validate effective and safe products that will ameliorate this issue. The presented experiment was properly designed and described. Some minor editorial comments are aimed to improve the readability of the article.

Comments:

Editorial note: please break the long chapters into shorter paragraphs, including one topic each, based on the general rule of drafting a paragraph. Reading 1-page long sections with a lot of random data is just not very pleasant

Abstract:

Line 35: „in group 3 group 4” – “and” is missing?

Line 35: should read significantly reduced?

Line 36: “All experimental groups treated showed..” one word too many?

Line 37: should read:  significant increase „in” SOD?

On the side note: when speaking about significant effect, it is common to show P value in the parentheses

Abstract is missing a sentence with conclusion/implication

Introduction:

Introduction is neatly drafted but it’s missing a link between EO and potential effects applied during heat stress. Please elaborate, so that you can formulate and justify the hypothesis.

Also, please justify in the introduction why the three particular treatments were selected and why in those combinations.

Materials and Methods:

Subchapter 2.2. Break the text into paragraphs for easier reading

Also, it would be nice addition to see the experimental timeline on a graph, including days, temperatures and sampling

Results:

Table 1. Why P-value is so low showing statistical significance on day0 prior to any treatment applied, when data show no differences between the groups (and correctly so!)? Please explain, and correct if necessary

Table 3. Impressive data, congratulations!

Discussion:

Some parts of the discussion (Lines 194-247; 296-306) contain some general information on heat stress and EO, which would fit better with the scope of introduction not discussion. The Authors should focus on their own data in the discussion. But, as I mentioned before, the Introduction is missing some valid data. So, maybe consider moving some information from discussion to introduction.

Line 260 – Tables are already mentioned in the Result section. No need to repeat the result, the Authors should discuss them and provide interpretation

Line 267 – define phytobiotics

Discussion is a little bit chaotic. Please, remove the paragraphs that should not be there and organize the remaining paragraphs so that they have a little bit better structure.

Author Response

Response to Reviewer 2 Comments

Dear Reviewer, #2, thank you very much for the time you have spent on reviewing our manuscript. Your comments are very valuable and helpful for revising our paper and guiding our research. We have studied those comments carefully and have made corrections, which we hope to meet with the approval. Revised portion in the new version were included and are highlighted in yellow in the reviewed manuscript. The following is our point-by-point response to reviewers’ comments:

Comments:

Editorial note: please break the long chapters into shorter paragraphs, including one topic each, based on the general rule of drafting a paragraph. Reading 1-page long sections with a lot of random data is just not very pleasant

Suggestion accepted.  Thank you.

Abstract:

Line 35: „in group 3 group 4” – “and” is missing?

Suggestion accepted.  Thank you.

Line 35: should read significantly reduced?

Suggestion accepted.  Thank you.

Line 36: “All experimental groups treated showed..” one word too many?

Suggestion accepted.  Thank you.

Line 37: should read:  significant increase „in” SOD?

Suggestion accepted.  Thank you.

On the side note: when speaking about significant effect, it is common to show P value in the parentheses

Suggestion accepted.  Thank you.

Abstract is missing a sentence with conclusion/implication

Suggestion accepted.  Thank you.

Introduction:

Introduction is neatly drafted but it’s missing a link between EO and potential effects applied during heat stress. Please elaborate, so that you can formulate and justify the hypothesis.

Suggestion accepted.  Thank you.

Also, please justify in the introduction why the three particular treatments were selected and why in those combinations.

Suggestion accepted. Thank you.

Materials and Methods:

Subchapter 2.2. Break the text into paragraphs for easier reading

Suggestion accepted.  Thank you.

Also, it would be nice addition to see the experimental timeline on a graph, including days, temperatures and sampling

Suggestion accepted.  Thank you.

Results:

Table 1. Why P-value is so low showing statistical significance on day0 prior to any treatment applied, when data show no differences between the groups (and correctly so!)? Please explain, and correct if necessary

Suggestion accepted.  This was a typo.  Thank you.

Table 3. Impressive data, congratulations!

Thank you.

Discussion:

Some parts of the discussion (Lines 194-247; 296-306) contain some general information on heat stress and EO, which would fit better with the scope of introduction not discussion. The Authors should focus on their own data in the discussion. But, as I mentioned before, the Introduction is missing some valid data. So, maybe consider moving some information from discussion to introduction.

Suggestion accepted.  Thank you.

Line 260 – Tables are already mentioned in the Result section. No need to repeat the result, the Authors should discuss them and provide interpretation

Suggestion accepted.  Thank you.

Line 267 – define phytobiotics

Suggestion accepted.  Thank you.

Discussion is a little bit chaotic. Please, remove the paragraphs that should not be there and organize the remaining paragraphs so that they have a little bit better structure.

Suggestion accepted.  Thank you.

Round 2

Reviewer 2 Report

The Reviewer appreciates that the Authors applied all the comments. Some revision is still necessary, mostly regarding Discussion. I would like to see a discussion more relating to the current study and results obtained, however avoiding repetitions. I know this is hard to draft a good discussion, but I would like to challenge the Authors to do so. There are plenty of reading materials on how to organize and draft a good discussion, to keep it complete, but still readable. This is just an example here, I am sure the Authors can find the best solution for their discussion

https://library.sacredheart.edu/c.php?g=29803&p=185933

At the moment there is a lot of good material in the discussion, but it doesn’t reflect the structure of the reverse pyramid. It is good to refer to the data, explain them and discuss with the literature. Please, avoid starting the paragraph randomly and out of blue. There should be a nice flow in the discussion. I would like to see that. Good luck!

Also, Experimental design is missing caption.

Author Response

The Reviewer appreciates that the Authors applied all the comments. Some revision is still necessary, mostly regarding Discussion. I would like to see a discussion more relating to the current study and results obtained, however avoiding repetitions. I know this is hard to draft a good discussion, but I would like to challenge the Authors to do so. There are plenty of reading materials on how to organize and draft a good discussion, to keep it complete, but still readable. This is just an example here, I am sure the Authors can find the best solution for their discussion

https://library.sacredheart.edu/c.php?g=29803&p=185933

Dear Reviewer, #1, thank you very much for the time you have spent on reviewing our manuscript. Your comments are very valuable and helpful for revising our paper and guiding our research. We have studied those comments carefully and have made corrections, which we hope to meet with the approval. The revised portion in the new version was included and is highlighted in blue in the reviewed manuscript.

At the moment there is a lot of good material in the discussion, but it doesn’t reflect the structure of the reverse pyramid. It is good to refer to the data, explain them, and discuss with the literature. Please, avoid starting the paragraph randomly and out of blue. There should be a nice flow in the discussion. I would like to see that. Good luck!

Suggestion accepted and discussion has been modified.  Thank you.

Also, Experimental design is missing caption.

Suggestion accepted. Thank you.
